# Identifying physical law of Hamiltonian systems via meta-learning

**Seungjun Lee, Haesang Yang, Woojae Seong**
Seoul National University
{tl7qns7ch,coupon3,wseong}@snu.ac.kr

## Abstract

Hamiltonian mechanics is an effective tool to represent many physical processes with concise yet well-generalized mathematical expressions. A well-modeled Hamiltonian makes it easy for researchers to analyze and forecast many related phenomena that are governed by the same physical law. However, in general, identifying a functional or shared expression of the Hamiltonian is very difficult. It requires carefully designed experiments and the researcher's insight that comes from years of experience. We propose that meta-learning algorithms can be potentially powerful data-driven tools for identifying the physical law governing Hamiltonian systems without any mathematical assumptions on the representation, but with observations from a set of systems governed by the same physical law. We show that a well meta-trained learner can identify the shared representation of the Hamiltonian by evaluating our method on several types of physical systems with various experimental settings.

## 1 Introduction

Hamiltonian mechanics, a reformulation of Newtonian mechanics, can be used to describe classical systems by focusing on modeling continuous-time evolution of system dynamics with a conservative quantity called Hamiltonian (Goldstein et al., 2002). Interestingly, the formalism of the Hamiltonian provides both geometrically meaningful interpretation (Arnol'd et al., 2001) and efficient numerical schemes (Feng & Qin, 2010) representing the state of complex systems in phase space with symplectic structure. Although formalism was originally developed for classical mechanics, it has been applied to various fields of physics, such as fluid mechanics (Salmon, 1988), statistical mechanics (Reichl, 1999), and quantum mechanics (Sakurai & Commins, 1995).

While it has many useful mathematical properties, establishing an appropriate Hamiltonian of the unknown phenomena is a challenging problem. A Hamiltonian for a system can be modeled by a shared expression of the Hamiltonian and physical parameters. For instance, the Hamiltonian of an ideal pendulum is described as $H = \frac{p^2}{2ml^2} + mgl(1 - \cos q)$ (shared expression), with mass $m$, pendulum length $l$, and gravity constant $g$ (physical parameters), whereas $q$ and $p$ are the angle of the pendulum and the corresponding conjugate momentum (state of the system), respectively. Once an appropriate functional of the Hamiltonian is established from observing several pendulums, a new pendulum-like system can be readily recognized by adapting new physical parameters on the expression. Therefore, identifying an appropriate expression of the Hamiltonian is an important yet extremely difficult problem in most science and engineering areas where there still remain numerous unknown processes where it is even uncertain whether a closed-form solution or mathematically clear expression exists.

In the recent era of deep learning, we can consider the use of learning-based algorithms to identify an appropriate expression of the Hamiltonian with sufficient data. To determine the Hamiltonian underlying the unknown physical process, the Hamiltonian should satisfy two fundamental conditions: (1) it should fit well on previously observed data or motions, (2) it should generalize well on newly observed data from new systems if the systems share the same physical law with previous ones. The first condition has been mitigated by explicitly incorporating symplectic structure or conservation laws on neural networks, called Hamiltonian neural networks (HNN) (Greydanus et al., 2019) for learning Hamiltonian dynamics. HNN and its variants have been shown to be effective in learning

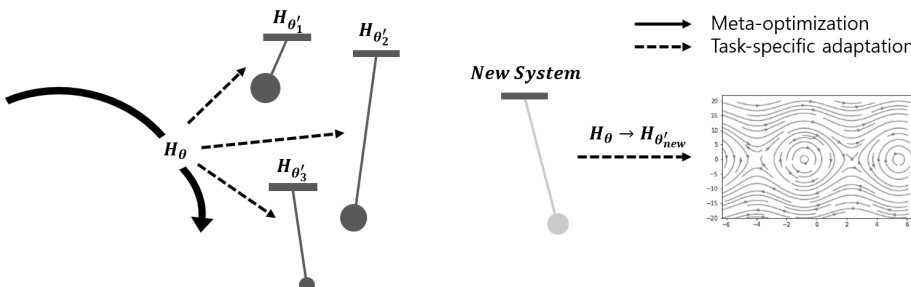

Figure 1: There is a resemblance between meta-learning and identifying the physical laws of Hamiltonian. A hypothesized governing equation of Hamiltonian, usually corrected and established by evaluating many related systems, could be learned using meta-learning as a data-driven method (left). Then, a well-established Hamiltonian can be utilized to predict new system dynamics, which could be viewed as a meta-transfer process by a well-trained meta-learner (right).

many useful properties of the Hamiltonian (Toth et al., 2020; Chen et al., 2020; Zhong et al., 2020a; Sanchez-Gonzalez et al., 2019; Jin et al., 2020). In their experiments, it has been shown that HNN and its variants work well on learning conservation laws or continuous-time translational symmetry, enable the learning of complex systems stably by incorporating numerical integrators and generalize on multiple initial conditions or controls for the given system. However, there is limited work regarding a trained model that works well on totally new systems governed by the same physical law with novel physical parameters.

To consider the second condition, we propose that meta-learning, which aims to train a model well generalized on novel data from observing a few examples, can be a potential key to learning a functional of Hamiltonian as a data-driven method. There have been several representative categories of meta-learning algorithms, such as the metric-based method (Snell et al., 2017; Sung et al., 2018), black-box method (Santoro et al., 2016; Bertinetto et al., 2019), and gradient-based method (Rusu et al., 2019; Flennerhag et al., 2020). Among these methods, we especially focus on the gradient-based method, which is readily compatible with any differentiable model and flexibly applicable to a wide variety of learning problems (Finn et al., 2017; Xu et al., 2018; Hospedales et al., 2020).

One of the most successful algorithms of the gradient-based method is Model-Agnostic Meta-Learning (MAML) (Finn et al., 2017), which consists of a task-specific adaptation process and a meta-optimization process. The key observations supporting its potential are the resemblance between these processes and the identification of the physical laws of the Hamiltonian. The schematic is shown in Figure 1. The task-adaptation process, which adapts the initial model parameters to a task-specific train set, resembles the process of adapting hypothesized governing equations to observations of several physical systems. The meta-optimization process, which updates the initial model parameters by validating each task-specific adapted parameters to a task-specific test set, is similar to correcting the hypothesized governing equations by validating each system-specific Hamiltonian on new data from the corresponding physical systems. In addition, (Raghu et al., 2020) proposed that the recent success behind these meta-learning algorithms was due to providing qualitative shared representation across tasks rather than learning initial model parameters that encourage rapid adaptation (Finn et al., 2017). This hypothesis may support our suggestion that a meta-learner can be efficient in identifying the shared representation of a Hamiltonian. From this point of view, we experiment on several types of physical systems to verify whether these meta-learning algorithms are beneficial to our desired learning problems. Our contributions are summarized as follows:

- We formulate the problem of identifying the shared representation of unknown Hamiltonian as a meta-learning problem.
- For learning to identify the Hamiltonian representations, we incorporate the HNN architecture on meta-learning algorithms.
- After meta-training the meta-learner, we adapt the model on new systems by learning the data of partial observations and predict the dynamics of the systems as a vector field in phase space.
- We evaluate our method on several types of physical systems to explore the efficiency of our methods with various experimental settings.

## 2 PRELIMINARIES

### 2.1 HAMILTONIAN NEURAL NETWORKS

In Hamiltonian mechanics, the state of a system can be described by the vector of canonical coordinates, $\boldsymbol{x} = (\boldsymbol{q}, \boldsymbol{p})$, which consist of position, $\boldsymbol{q} = (q_1, q_2, ..., q_n)$ and its conjugate momentum, $\boldsymbol{p} = (p_1, p_2, ..., p_n)$ in phase space, where $n$ is the degree of freedom of the system. Then, the time evolution of the system is governed by Hamilton's equations $\dot{\boldsymbol{x}} = \left( \frac{\partial H}{\partial \boldsymbol{p}}, -\frac{\partial H}{\partial \boldsymbol{q}} \right) = \Omega \nabla_{\boldsymbol{x}} H(\boldsymbol{x})$, where

$H(\boldsymbol{x}) : \mathbb{R}^{2n} \to \mathbb{R}$ is the Hamiltonian that is conservative during the process and $\Omega = \begin{bmatrix} 0 & I_n \\ -I_n & 0 \end{bmatrix}$ is

a $2n \times 2n$ skew-symmetric matrix. From the Hamiltonian equations, the Hamiltonian vector field in phase space, which is interpreted as the time evolution of the system $\dot{\boldsymbol{x}}$, is the symplectic gradient of the Hamiltonian $\Omega \nabla_{\boldsymbol{x}} H(\boldsymbol{x})$, which is determined by the Hamiltonian function and the state of the system itself. Then, the trajectory of the state can be computed by integrating the symplectic gradient of the Hamiltonian. If the Hamiltonian does not depend on the time variable, the Hamiltonian remains constant during the time evolution, because moving along the direction of the symplectic gradient keeps the Hamiltonian constant (Arnol'd, 2013). In (Greydanus et al., 2019), the Hamiltonian function can be approximated by neural networks, $H_\theta$, called HNN. To make the Hamiltonian function constant in motion, the loss of HNN is defined by the distance between the true vector field and the symplectic gradient of $H_\theta$,

$$\mathcal{L}_{HNN} = \| \dot{\boldsymbol{x}} - \Omega \nabla_{\boldsymbol{x}} H_\theta(\boldsymbol{x}) \|_2^2. \tag{1}$$

### 2.2 MODEL-AGNOSTIC META-LEARNING AND FEATURE REUSE HYPOTHESES

A key assumption behind MAML is that separately trained models for each task share meta-initial parameters $\theta$ that could be improved rapidly for any task (Finn et al., 2017). Suppose that each given task, $\mathcal{T}_i$, composed of $\mathcal{D}_i = \{\mathcal{D}_i^{tr}, \mathcal{D}_i^{te}\}$, is drawn from a task distribution, $\mathcal{T}_i \sim p(\mathcal{T})$. The learning algorithms usually consist of bi-level optimization processes; (inner-loop) the task-specific adaptation to each train set,

$$\theta_i' = \theta - \alpha \nabla_\theta \mathcal{L}_{\mathcal{T}_i} \left( \mathcal{D}_i^{tr}; \theta \right), \tag{2}$$

and (outer-loop) the meta-optimization on each test set,

$$\theta \leftarrow \theta - \beta \nabla_\theta \sum_{\mathcal{T}_i \sim p(\mathcal{T})} \mathcal{L}_{\mathcal{T}_i}(\mathcal{D}_i^{te}; \theta_i'), \tag{3}$$

where $\theta$ could be any differentiable model's parameters that are expected to learn the shared representations of various tasks, and $\alpha$ and $\beta$ are the step sizes of the inner and outer loops, respectively.

Meanwhile, (Raghu et al., 2020) observed that during the inner-loop process, the task-specific distinction of the model parameters $\theta$ is mostly from the last layer of the networks, whereas the entire body of the model hardly changed. Therefore, they hypothesized that the body of the model behaves as a shared representation across the different tasks, whereas the head of the model behaves as a task-specific parameter, which is called the *feature reuse* hypothesis. From this hypothesis, they proposed a gradient-based meta-learning algorithm called Almost No Inner Loop (ANIL) by slightly modifying MAML by freezing all but updating the last layer of the networks during the inner-loop process. They showed that ANIL performs on par or better than MAML on several benchmarks, and has a computational benefit compared to its counterpart. For the algorithm, when the meta-learner consists of $l$ layers $\theta = (\theta^{(1)}, ..., \theta^{(l-1)}, \theta^{(l)})$, the inner-loop update is modified as

$$\theta_i' = (\theta^{(1)}, ..., \theta^{(l-1)}, \theta^{(l)} - \alpha \nabla_{\theta^{(l)}} \mathcal{L}_{\mathcal{T}_i} \left( \mathcal{D}_i^{tr}; \theta \right)). \tag{4}$$

As many physical processes could be expressed as an invariant shared expression of Hamiltonian and variable physical parameters, such meta-learning scheme, which encourages to separate the invariant part and varying part, can be expected to be more efficient to learn new systems by the relatively small number of parameter update.

## 3 METHOD

### 3.1 IDENTIFYING SHARED REPRESENTATION OF HAMILTONIAN VIA META-LEARNER

The main goal of our study is to train a model to identify the shared representation of the Hamiltonian using observations of dynamics from several systems that are assumed to be governed by the same physical law with different physical parameters. From a meta-learning point of view, each system is regarded as a task $\mathcal{T}_i$, where the physical parameters of the system are drawn from the distribution of $p(\mathcal{T})$. The observations of the system $\mathcal{T}_i$ can be split into $\mathcal{D}_i = \{\mathcal{D}_i^{tr}, \mathcal{D}_i^{te}\}$, where $\mathcal{D}_i^{tr}$ and $\mathcal{D}_i^{te}$ denote the task-specific train and test sets, respectively. The observations of both $\mathcal{D}_i^{tr}$ and $\mathcal{D}_i^{te}$ are given by a set of tuples of canonical coordinates $\boldsymbol{x} = (\boldsymbol{q}, \boldsymbol{p})$ and their time-derivatives $\dot{\boldsymbol{x}} = (\dot{\boldsymbol{q}}, \dot{\boldsymbol{p}})$ as the ground truth.

For each system, the task-specific model parameters are obtained from Equation 2 or Equation 4 by computing the task-specific loss using Equation 1 on each train set $\mathcal{D}_i^{tr}$,

$$\mathcal{L}_{\mathcal{T}_i}(\mathcal{D}_i^{tr}; \theta) = \sum_{(\boldsymbol{x}, \dot{\boldsymbol{x}}) \sim \mathcal{D}_i^{tr}} \left\| \dot{\boldsymbol{x}} - \Omega \nabla_{\boldsymbol{x}} H_\theta(\boldsymbol{x}) \right\|_2^2, \tag{5}$$

and the meta-optimization can be operated on the batch of systems as Equation 3 by minimizing the loss over the batch of physical parameters sampled from $p(\mathcal{T})$. Each loss is computed by evaluating each task-specific adapted model parameters $\theta_i'$ to each test set $\mathcal{D}_i^{tr}$,

$$\sum_{\mathcal{T}_i \sim p(\mathcal{T})} \mathcal{L}_{\mathcal{T}_i}(\mathcal{D}_i^{te}; \theta_i') = \sum_{\mathcal{T}_i \sim p(\mathcal{T})} \sum_{(\boldsymbol{x}, \dot{\boldsymbol{x}}) \sim \mathcal{D}_i^{te}} \left\| \dot{\boldsymbol{x}} - \Omega \nabla_{\boldsymbol{x}} H_{\theta_i'}(\boldsymbol{x}) \right\|_2^2. \tag{6}$$

Depending on the inner-loop methods, we call the algorithms Hamiltonian Model-Agnostic Meta-Learning (HAMAML) when using Equation 2, and Hamiltonian Almost No Inner-Loop (HANIL) when using Equation 4.

### 3.2 LEARNING NEW SYSTEM DYNAMICS FROM PARTIAL OBSERVATIONS

It can be expected that if the learner is efficient in identifying the underlying physical nature of an unknown process, the model can appropriately predict the dynamics of novel systems from their partial observations. To evaluate whether a learner is efficient in identifying the representation of an unknown Hamiltonian, we set up the following validating scheme on several types of physical systems. After meta-training, novel systems are given as meta-test sets $\mathcal{D}_{new} = \{\mathcal{D}_{new}^{tr}, \mathcal{D}_{new}^{te}\}$ generated from the system's Hamiltonian with novel physical parameters $\mathcal{T}_{new} \sim p(\mathcal{T})$. The train set $\mathcal{D}_{new}^{tr}$ is used to adapt the learner on the new system dynamics ($\theta \rightarrow \theta_{new}'$). The new observed systems are given as $K$ trajectories, which are reminiscent settings of the $K$-shot learning problem (Snell et al., 2017). Starting from the $K$ initial states, each trajectory is obtained by integrating the time-derivatives of the systems from 0s to $T$s, with sampling rate $\frac{L}{T}$ which yields $L$ rolled out sequence of states. The test set $\mathcal{D}_{new}^{te}$ is for validating the performance of the adapted model $\theta_{new}'$.

## 4 RELATED WORKS

### 4.1 LEARNING DYNAMICS WITH NEURAL NETWORKS

Several works on learning dynamical systems usually use various forms of graph neural networks (Battaglia et al., 2016; Santoro et al., 2017; Sanchez-Gonzalez et al., 2018; Kipf et al., 2018; Battaglia et al., 2018; Sanchez-Gonzalez et al., 2020), where each node usually represents the state of individual objects and each edge between two arbitrary nodes usually represents the relation of the nodes. (Chen et al., 2018) introduced a new type of deep architecture, which is considered as the differentiable ordinary differential equations (ODE) solver parametrized by neural networks, called neural ODEs. The most related to our work, (Greydanus et al., 2019) introduced Hamiltonian Neural Networks (HNNs) to learn the dynamics of Hamiltonian systems by parameterizing the Hamiltonian with neural networks. The HNNs have been developed through combining graph networks for learning interacting systems (Sanchez-Gonzalez et al., 2019) or generative networks for learning Hamiltonian from high-dimensional data (Toth et al., 2020). (Chen et al., 2020) proposed Symplectic recurrent neural networks to handle observed trajectories of complex Hamiltonian

systems by incorporating leapfrog integrator to recurrent neural networks. (Zhong et al., 2020a) considered additional control terms to learn the conservative Hamiltonian dynamics and the dissipative Hamiltonian dynamics (Zhong et al., 2020b). For learning Hamiltonian in an intrinsic way, (Jin et al., 2020) proposed Symplectic networks where the architecture consists of the compositional modules for preserving the symplectic structure. In most existing studies for learning Hamiltonian, one model is trained per one system, and the evaluations are restricted to the same physical system during training with new initial conditions. Our focus is learning the shared representation of Hamiltonian which can be reused to predict new related systems governed by the same physical law that appear throughout the previously observed systems.

## 4.2 IDENTIFYING PHYSICAL LAWS OR GOVERNING EQUATIONS FROM DATA

In earlier works, symbolic regression was utilized to search the mathematical expressions of the governing equations automatically from observed data (Bongard & Lipson, 2007; Schmidt & Lipson, 2009). These methods have been developed by constructing a dictionary of candidate nonlinear functions (Brunton et al., 2016; Mangan et al., 2017) or partial differentiations (Rudy et al., 2017; Schaeffer, 2017) by incorporating a sparsity-promoting algorithm to extract the governing equations. Recently, they have been refined through combining with neural networks (Udrescu & Tegmark, 2020; Both et al., 2019; Atkinson et al., 2019; Sahoo et al., 2018) or graph structures (Cranmer et al., 2019; 2020). In contrast to our work, the existing methods of identifying the governing representations used the symbolic representation underlying the assumptions that the unknown physical laws are expressed by combinations of known mathematical terms.

## 4.3 GRADIENT-BASED META-LEARNING

The core learning method of our work is related to gradient-based meta-learning algorithms (Finn et al., 2017). The learning method has been developed by training additional learning rate (Li et al., 2017), removing the second-order derivatives (Nichol et al., 2018) and stabilizing the training procedure (Antoniou et al., 2019). Meanwhile, in another direction of the research, the model parameters are separated by the shared representation part which is mainly updated in the outer-loop and task-specific varying part which is mainly updated in the inner-loop (Lee & Choi, 2018; Javed & White, 2019; Flennerhag et al., 2020; Raghu et al., 2020; Zhou et al., 2020). Our study mainly focuses on whether a meta-learning could identify the shared expression of the Hamiltonian from several observed systems. MAML is chosen as the representative of the meta-learning algorithm. Also, for verifying whether the separative learning scheme improves the model to identify the shared expression of the Hamiltonian, we choose ANIL as the representative of the meta-learning with the separative learning scheme. Since, without any structural difference, it is sufficient to verify that the existence of the separative learning scheme would be beneficial.

## 5 EXPERIMENTS

### 5.1 TYPES OF PHYSICAL SYSTEMS

**Spring-Mass.** Hamiltonian of the system is described by $H = \frac{p^2}{2m} + \frac{k(q-q_0)^2}{2}$, where the physical parameters $m$, $k$, and $q_0$ are the mass, spring constant, and equilibrium position, respectively. $q$ and $p$ are the position and conjugate momentum of the system, respectively.

**Pendulum.** Hamiltonian of the system is described by $H = \frac{p^2}{2ml^2} + mgl(1 - \cos(q - q_0))$, where the physical parameters $m$, $l$, and $q_0$ are the mass, pendulum length, and equilibrium angle from the vertical, respectively. $q$ and $p$ are the pendulum angle from the vertical and conjugate momentum of the system, respectively. $g$ denotes the gravitational acceleration.

**Kepler problem.** The system consists of two objects that are attracted to each other by gravitational force. Hamiltonian of the system is described by $H = \frac{|\boldsymbol{p}|^2}{2m} - \frac{GMm}{|\boldsymbol{q}-\boldsymbol{q}_0|}$, where the physical parameters $M$ and $m$ are the mass of the two bodies and we set the object of $M$ to be stationary at $\boldsymbol{q}_0 = (q_x, q_y)$ of the coordinate system. The state is represented by $\boldsymbol{q} = (q_1, q_2)$ and $\boldsymbol{p} = (p_1, p_2)$ which denote the position and conjugate momentum of the object $m$ in two-dimensional space, respectively. $G$ denotes the gravitational constant.

Table 1: MSEs across the test sets of 10 new systems from adapting to the corresponding train sets after 10 gradient steps.

| Systems | Learner | Observations | | | |
| | | Point Dynamics | | Trajectories | |
| | | 25-shot | 50-shot | 5-shot | 10-shot |
|---|---|---|---|---|---|
| Spring-Mass | HNN from Scratch | 82.3±48.9 | 82.1±48.6 | 84.7±48.9 | 82.4±48.8 |
| | Pretrained HNN | 8.2±13.3 | 4.9±6.6 | 11.9±14.1 | 6.6±10.0 |
| | Naive NN + MAML | 159.8±65.6 | 155.8±64.4 | 195.7±87.1 | 162.3±66.5 |
| | Naive NN + ANIL | 16.6±15.7 | 16.2±13.6 | 36.0±25.4 | 19.5±17.4 |
| | HAMAML | 2.9±2.4 | 2.4±1.4 | 4.2±5.2 | 1.6±0.6 |
| | HANIL | **0.02±0.01** | **0.01±0.006** | **0.4±0.4** | **0.1±0.07** |
| Pendulum | HNN from Scratch | 6.5±2.6 | 6.1±2.1 | 6.7±2.1 | 6.6±1.9 |
| | Pretrained HNN | 5.4±3.5 | 5.1±3.1 | 5.9±3.5 | 5.8±3.2 |
| | Naive NN + MAML | 5.8±3.7 | 5.3±3.3 | 10.3±8.1 | 8.3±4.5 |
| | Naive NN + ANIL | 2.3±1.8 | 2.0±1.4 | 3.3±2.4 | 2.8±1.8 |
| | HAMAML | 1.5±0.5 | 1.4±0.3 | 4.7±2.9 | 2.9±1.9 |
| | HANIL | **0.02±0.04** | **0.003±0.001** | **0.6±1.2** | **0.04±0.02** |
| Kepler | HNN from Scratch | 6.7±11.3 | 6.6±11.3 | 7.4±12.4 | 6.9±11.6 |
| | Pretrained HNN | 1.7±2.6 | 1.6±2.6 | 1.9±2.9 | 1.7±2.7 |
| | Naive NN + MAML | 1.5±1.5 | 0.9±0.4 | 2.8±4.6 | 1.3±0.7 |
| | Navie NN + ANIL | 1.1±0.9 | 0.9±0.6 | 2.4±3.3 | 1.2±0.9 |
| | HAMAML | 1.5±1.5 | 1.3±1.6 | 1.7±2.4 | 1.5±1.6 |
| | HANIL | **0.33±0.19** | **0.33±0.18** | **0.33±0.19** | **0.33±0.18** |

## 5.2 EXPERIMENTAL SETTINGS

**Datasets.** During the meta-training, we generate 10,000 tasks for meta-train sets of all systems. Each meta-train set consists of task-specific train set and test set given by 50 randomly sampled point states $x$ and their time-derivatives $\dot{x}$ in phase space with task-specific physical parameters. The distributions of the sampled states and physical parameters for each physical process are described in Appendix A.1. During the meta-testing, the distributions of sampled states and physical parameters are the same as in the meta-training stage. To evaluate learners on the efficacy of identifying Hamiltonian of the new systems, meta-test sets are constructed as the following ways.

(1) Observing the new systems as point dynamics in phase space: $\mathcal{D}_{new}^{tr}$ consists of randomly sampled points in phase space with $K = \{25, 50\}$ and $L = 1$, and $\mathcal{D}_{new}^{te}$ consists of equally fine-spaced points in phase space.

(2) Observing the new systems as trajectories in phase space: $\mathcal{D}_{new}^{tr}$ consists of randomly sampled trajectories and $\mathcal{D}_{new}^{te}$ consists of equally fine-spaced points in phase space. The number of sampled trajectories with $K = \{5, 10\}$ and $L = 5$ sequences during $T = 1s$.

The trajectories are obtained by integrating the symplectic gradient of the Hamiltonian from initial states using the adaptive step-size Runge-Kutta method (Hairer et al., 1993). Fine-spaced test sets consist of equally spaced grids for each coordinate in the region of the phase space where we sampled the point states. More details about data sets are represented in the Appendix A.1.

**Baselines.** We took several learners as baselines to assess the efficacy of our proposed methods, such as (1) training HNN on $\mathcal{D}_{new}^{tr}$ from scratch (random initialization), (2) pretrained HNN across all of the meta-train set, (3) meta-trained naive fully connected neural networks (Naive NN), which are given the inputs $x$ and the outputs $\dot{x}$ with MAML, and (4) with ANIL. The architectures and training details of the baselines and our methods are represented in the Appendix A.2.

**Evaluations.** The evaluation metric is the average of the mean squared errors (MSE) between true vector fields $\dot{x}$ and the symplectic gradients of predicted Hamiltonian $\Omega \nabla_x H_\theta(x)$ across the test sets of new systems $\mathcal{D}_{new}^{te}$ by adapting the learners to the corresponding train set of the new systems $\mathcal{D}_{new}^{tr}$. For all types of systems, the averaged MSE of randomly sampled 10 new systems are averaged for evaluating the learners after 10 gradient steps adapting to each $\mathcal{D}_{new}^{tr}$. We also predict the state trajectories and the corresponding energies by integrating the output vector fields of the learners adapted to a new system by observing samples of point dynamics. Following (Greydanus et al., 2019), we evaluate the MSEs of the predicted trajectories and energies from their corresponding ground truth at each time step. The predicted values are computed by the learners adapted to 50 randomly sampled point dynamics of new systems in phase space after 50 gradient steps.

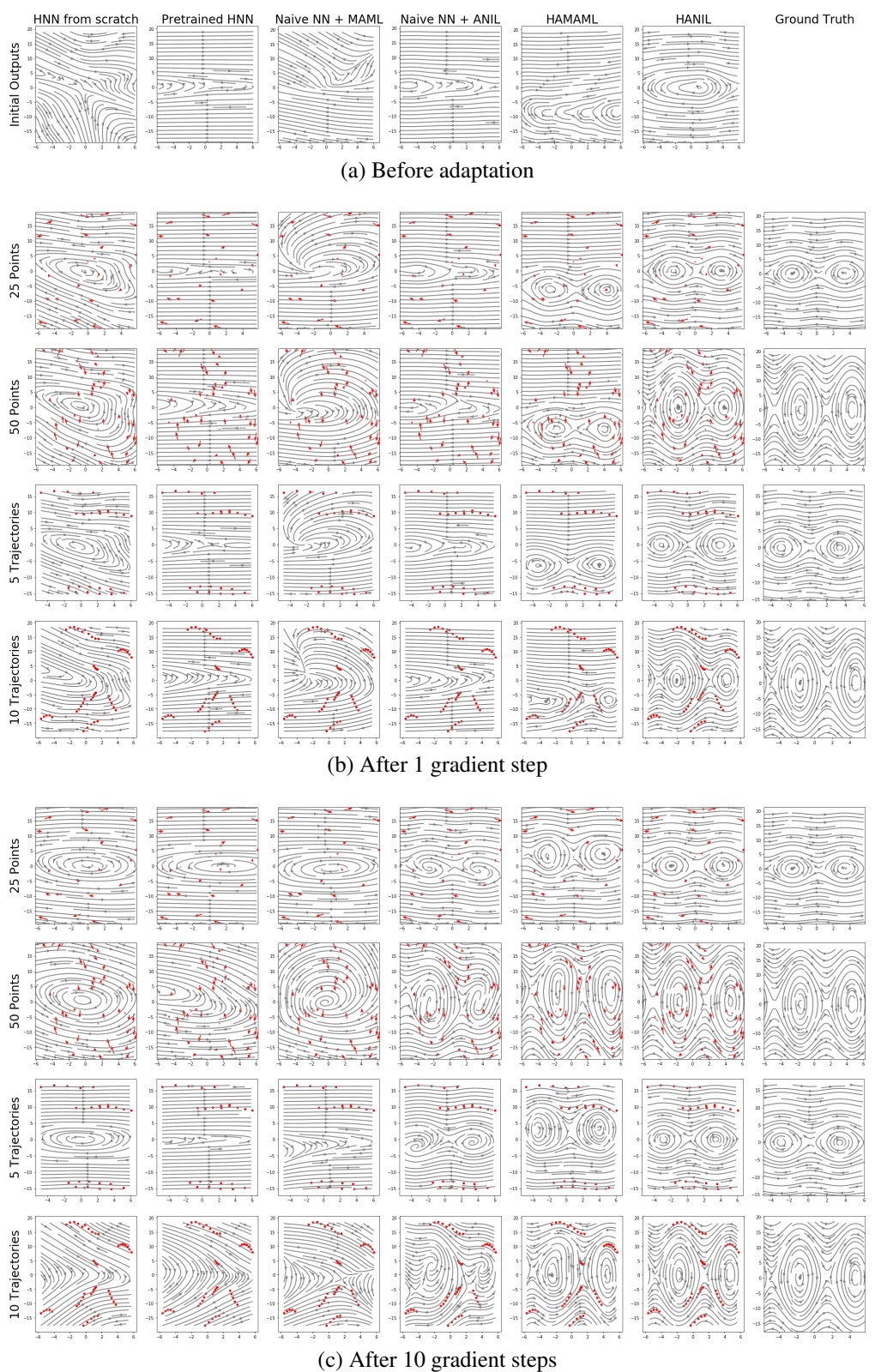

Figure 2: Predicted vector fields (gray streamlines) by adapting the learners to observations of new pendulum systems given as point dynamics (red arrows) or trajectories (red dots) after the corresponding gradient steps. The $x$-axis and $y$-axis denote $q$ and $p$, respectively.

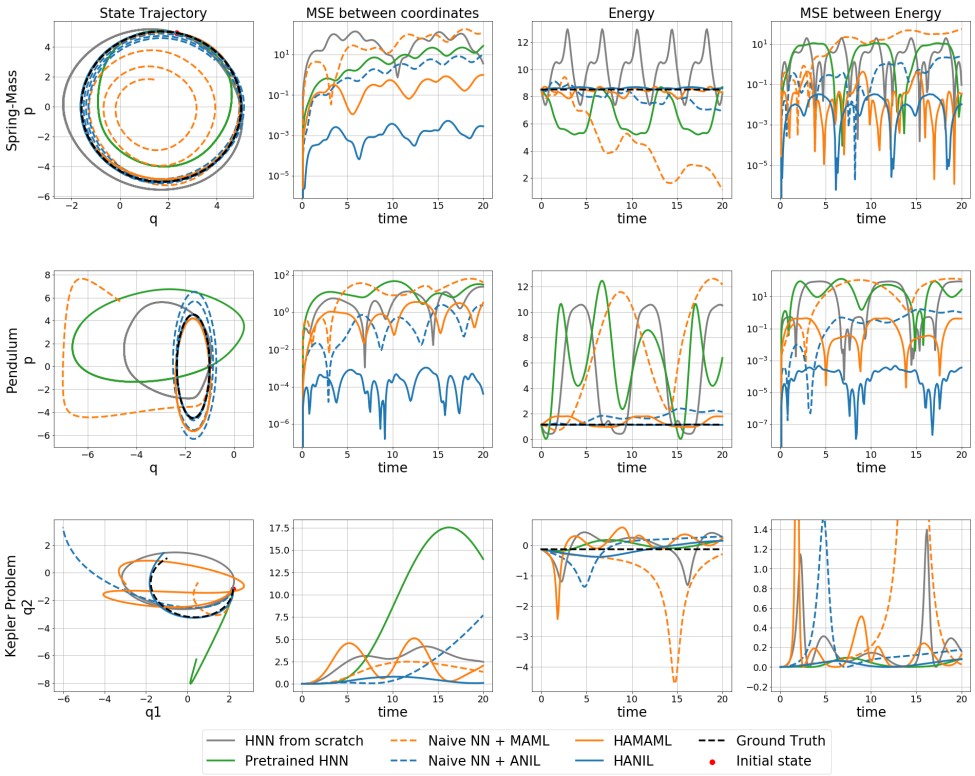

Figure 3: Predicted state trajectories (or position coordinates for Kepler problem) and the corresponding energies, and the corresponding MSEs at each time step. Note that the MSEs of spring-mass and pendulum are represented as log-scaled.

## 5.3 RESULTS

**Quantitative results.** The quantitative results of meta-testing performance are shown in Table 1. It is shown that HANIL outperforms the others for all experimental settings in all types of physical systems. When observing 25-shot point dynamics and 5-shot trajectories, the number of given samples in the phase space is the same, and the same is true for observing 50-shot point dynamics and 10-shot trajectories ($L = 5$). Comparing the point dynamics and trajectories with the same number of given samples, learning from observing point dynamics is slightly more accurate than that from observing trajectories.

**Predicted vector fields.** In Figure 2, predicted pendulum dynamics by adapting the learners from observing partial observations are represented as phase portraits by the corresponding gradient steps. Note that those of spring-mass systems are shown in Figure 5 in Appendix A.3. In Figure 2 (a), the initial outputs of vector fields from the learners are represented. During the adaptation to the given observations, the output vectors of each learner are evolved to fit on the observations based on their own prior belief or representation learned from the meta-train set. In detail, HNN from scratch fails to predict the dynamics of new systems from partial observations. At least hundreds of states should be given as train set, and thousands of gradient steps are required for training HNN for learning a system (Greydanus et al., 2019). However, in our meta-testing, up to 50 states are given to adapt to the new system with few gradient steps. Thus, the number of samples and gradient steps is too small to train HNN without any inductive bias. Pretrained HNN also fails, even though it is trained using the meta-train sets. A model simply pretrained across all tasks may output the averaged values of time-derivative at each state point. As the time-derivatives of each state would varying sensitive to the physical parameters of the systems, the simple averaged values are likely to have very different patterns from the actual vector fields. Therefore, such pretrained model would not be efficient to learn appropriate shared representation across the systems. Naive NNs, with MAML and ANIL also fail to predict the dynamics because naive NNs are hard to grasp the continuous and conservative structure of the vector fields where the number of given data are not sufficient to adapt to new

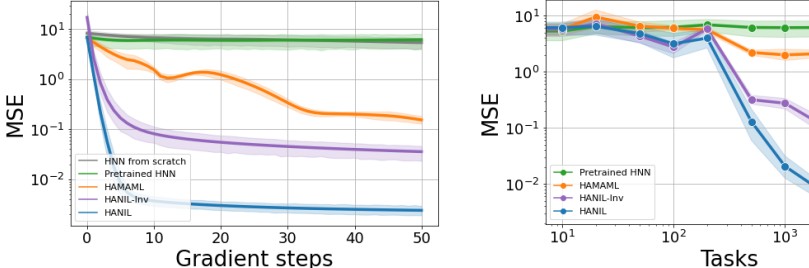

Figure 4: Ablation studies conducted on 10 new pendulum systems by comparing (a) Learning curves during task-adaptation, and (b) the effects of the size of the meta-train sets.

systems. Thus, the phase portraits of them are discontinuous or dissipative-like shape. HANIL can accurately predict the dynamics of new systems from partial observations with few gradient steps, while HAMAML is slower than HANIL to adapt the true vector fields because of the larger number of parameters to update in the adaptation process.

**Trajectory predictions.** In Figure 3, we also evaluate the learners adapted to the new systems through their predictions of state and the corresponding energies starting from the initial states during $20s$. HANIL (blue lines) adapted to new systems predicts the states and energy trajectories with relatively small errors from the ground truth (black dashed lines), whereas the others fail to predict the right trajectories and energies of the system at each time step.

**Ablation study.** We conduct ablation studies to verify the efficacy of the separative learning schemes by comparing the learning curves during task-adaptation, and the effects of the size of the meta-train sets. We evaluate the same evaluation of MSEs as described in Section 5.2 but varying gradient steps from 0 to 50 to see the learning curves and varying the number of tasks from 10 to 10,000 which are observed during meta-training to verify the effects of the size of the meta-train set. In addition, in order to see how the number of parameters updated during the inner-loop affects the performance, we add another comparison meta-learned learner called HANIL-Inverse (HANIL-INV). The method updates all but except the first layer of the network during the inner-loop, while HANIL only updates the last layer of the network. Therefore, during the inner-loop, the number of updated parameters of the HANIL-INV is between those of the HAMAML and HANIL. In Figure 4, the results of ablation studies on pendulum systems are represented where the dynamics of new systems are given as 50 point dynamics. HANIL-Inv converges to the error between HAMAML and HANIL, while the HNN from scratch and the pretrained HNN are hardly improved in both studies. As comparing the meta-trained learners with the others, meta-learning improves the performance to predict new systems through their ability to learn the shared representation. As comparing HAMAML and meta-trained learners with the separative learning scheme such as HANIL-INV and HANIL, it seems beneficial to separately learn the shared representation and physical parameters through the separative learning scheme. In addition, since the updated parameters are limited to the last layer during the inner-loop, HANIL converges with the lowest errors, which would be most efficient to generalize the related systems with the same physical law, as fewer updated parameters are required for the new system. Meanwhile, in Figure 4 (b), the point where the effect of the size of tasks to the meta-trained learners begins to be noticeable is between 200 and 500 and slowly converges at around 10,000.

## 6 CONCLUSIONS

By observing the resemblance between seemingly unrelated problems, identifying the Hamiltonian and meta-learning, we formulate the problem of identifying the Hamiltonian as a meta-learning problem. We incorporate HNN, which is an efficient architecture for learning Hamiltonian, with meta-learning algorithms in order to discover the shared representation of unknown Hamiltonian across observed physical systems. Comparing the baseline models with various experiments, we show that our proposed methods, especially HANIL, is efficient to learn totally new systems dynamics governed by the same underlying physical laws. The results state that our proposed methods have the ability to extract the meta-transferable knowledge, which can be considered as physical nature across the observed physical systems during meta-training.

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

# A APPENDIX

## A.1 SYSTEM DETAILS

**Spring-Mass.** The physical parameters are randomly sampled from $(m, k, q_0) \in ([0.5, 5], [0.5, 5], [-5, 5])$. The initial states are randomly sampled from $(q, p) \in ([-10, 10]^2)$. Fine-spaced test sets of new systems consist of 50 equally spaced grids for each coordinate in the region of the phase space where we sampled the point states. Therefore, there are 2,500 grids points in the test sets.

**Pendulum.** The physical parameters are randomly sampled from $(m, l, q_0) \in ([0.5, 5], [0.5, 5], [-\pi, \pi])$. We fix the gravitational acceleration as $g = 1$. The initial states are randomly sampled from $(q, p) \in ([-2\pi, 2\pi], [-20, 20])$. Fine-spaced test sets of new systems consist of 50 equally spaced grids for each coordinate in the region of the phase space where we sampled the point states. Therefore, there are 2,500 grids points in the test sets.

**Kepler Problem.** The physical parameters are randomly sampled from $(M, m, q_x, q_y) \in ([0.5, 2.5]^2, [-2.5, 2.5]^2)$. We fix the gravitational constant as $G = 1$. The initial states are randomly sampled from $(\boldsymbol{q}, \boldsymbol{p}) \in ([-5, 5]^4)$. Fine-spaced test sets of new systems consist of 10 equally spaced grids for each coordinate in the region of the phase space where we sampled the point states. Therefore, there are 10,000 grids points in the test sets.

## A.2 IMPLEMENTATION DETAILS

For all tasks, we took the baseline model as fully connected neural networks with the size of state dimensions - 64 Softplus - 64 Softplus - 64 Softplus - state dimensions and the HNN model as fully connected neural networks with the size of state dimensions - 64 Softplus - 64 Softplus - 64 Softplus - 1 dimension. We searched the hyperparameters, exploring the size of hidden dimensions from $\{32, 64, 128, 256\}$, the number of layers from $\{1, 2, 3, 4\}$, and the activation function from $\{\text{Sigmoid}, \text{Tanh}, \text{Relu}, \text{Softplus}\}$. During meta-training or pretraining, we use the Adam optimizer (Kingma & Ba, 2015) on outer-loop with learning rate of 0.001 and use gradient descent on inner-loop with learning rate of 0.002. For all systems, we set the number of task batches of 10, inner gradient updates of 5, and episodes of outer loop of 100 for meta-optimization. During the meta-testing, we also use the Adam optimizer with a learning rate 0.002. There is no weight decay for all.

## A.3 ADDITIONAL RESULTS

More results and video could be available at `https://github.com/7tl7qns7ch/Identifying-Physical-Law`.

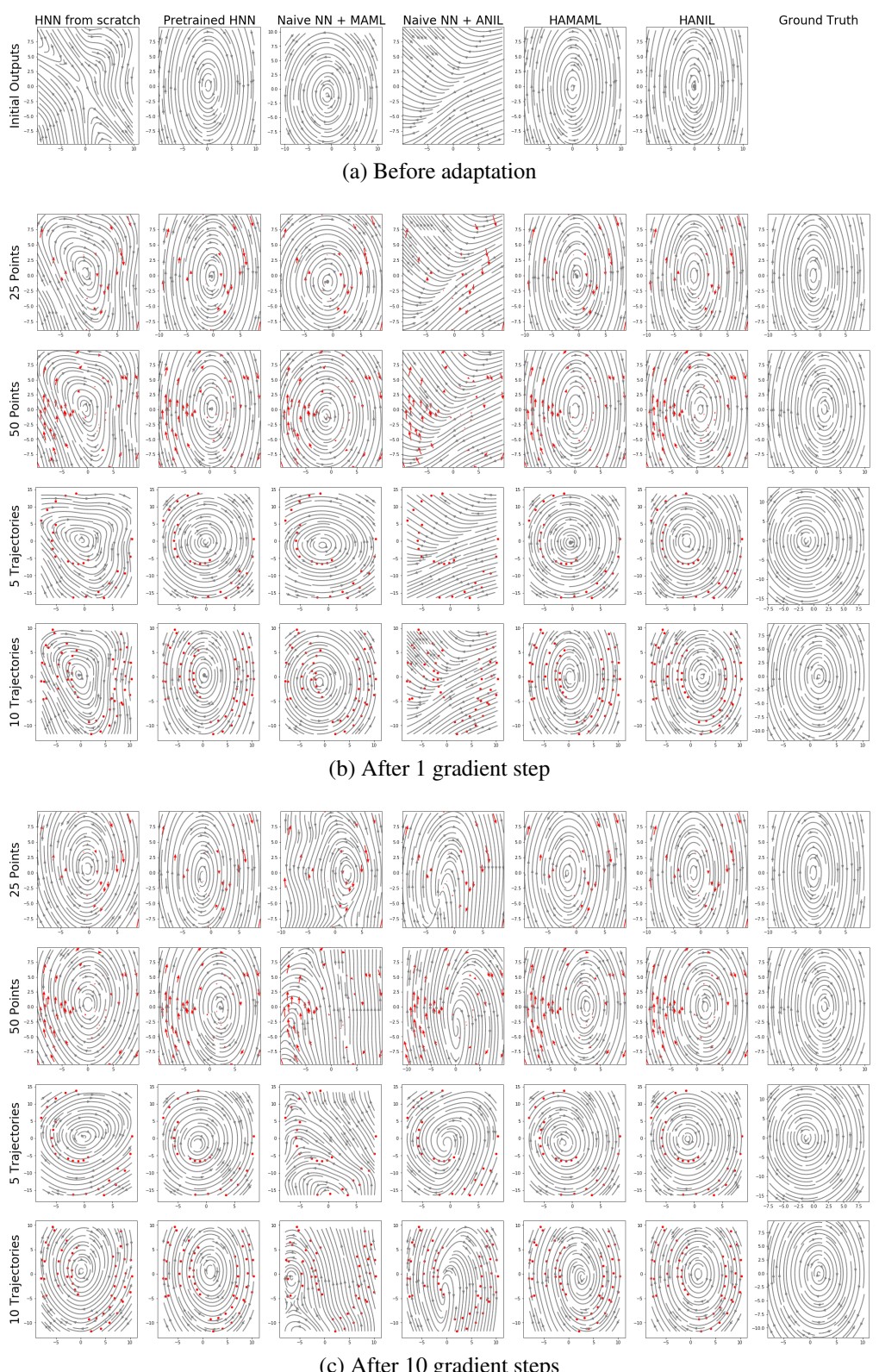

Figure 5: Predicted vector fields (gray streamlines) by adapting the learners to observations of new spring-mass systems given as point dynamics (red arrows) or trajectories (red dots) after the corresponding gradient steps. The $x$-axis and $y$-axis denote $q$ and $p$, respectively.

