# OpenReview forum: "Identifying Physical Law of Hamiltonian Systems via Meta-Learning"
_ICLR.cc/2021/Conference — ICLR 2021 Poster_

### Official Review · AnonReviewer1 · 2020-10-27
**Combination of HNNs and MAML/NIL**

**Rating:** 6
**Confidence:** 4

**Review:**

The authors combine Hamiltonian neural networks (HNN) with the MAML and ANIL neural learning approaches for Meta-Learning. The results are mostly consistent with what would be expected when combining them.
The paper is easy to follow and nice to read, and seems to have sufficient implementation details for reproducibility.

My main criticism is that there is not really novel technical content: the model is a trivial combination of MAML/NIL with HNNs. This could be ok in the following scenarios:
* If results were given on well-known important domains achieving state of the art. Currently all domains are very toy.
* If the paper gave a hint of what real-world problem settings could benefit from this specific combination, which does not seem to be part of the discussion.
* If the paper did a very thorough empirical investigation. The results are nice (I really like Figure 2), but otherwise none of the insights are really new, but just the union of the conclusions of the MAML/NIL papers and the HNN paper.

but those are not really well covered.

Probably the biggest value of the paper is that it brings together the HNN community (often closer to natural sciences communities) to the meta-learning community (often closer to data-science communities), and this paper may be a gateway from people from the former one to learn more about meta-learning, but I am not if this is sufficient to grant acceptance. I would not oppose an acceptance decision if the AC, or other reviewers decide to accept it on these grounds in its current form.

Additional comments:

I am a bit surprised HANIL is so much better than HAMAML, considering Naive ANIL is not so much better than Naive MAML comparatively (specially in SpringMass and and Pendulum point dynamics).

Was the number of gradient steps during evaluation fine-tuned for each baseline separately? Could this be the reason why MAML is comparatively worse? This is hinted in “while HAMAML is slower than HANIL to adapt the true vector fields because of the larger number of parameters to update in adaptation process”, but it would be nicer to see more supporting evidence or additional investigations.

All of the results in Table 1 corresponds to the evaluation of the loss of the derivatives, and not the error over an integrated trajectory right? I find it a bit unintuitive that the Naive network is on pair with the model on Kepler, but not on the other two domains. I would expect most of the advantage of the Hamiltonian model would come from the stability when integrating over longer trajectories, but not so noticeable on the predictions of gradients at single points. In fact in Table 1 from the original HNNs paper, they precisely show that the differences between the Baseline and the HNN are mostly w.r.t. preservation of energy, but that the losses on the derivatives are nearly identical.

It would be nice to see some experiments where the functional form of the Hamiltonian itself (rather than just hamiltonian parameters change across time). The title, introduction and related work, somehow makes it look like different physical laws will be inferred during the meta learning stage, however in practice the meta-learning task covers the same physical laws than those seen during training with different parameters. Otherwise the title is highly misleading, since it does not identify physical laws, but to identify specific instances of the same physical law.

How were the trajectories from Figure 3 chosen, fully at random?

Minor comments/typos (have not affected my decision):

Table 1: Possibly bold font for best model, or even better, some bar plots would make it easier to visualize the data.

weird grammar: “with sufficient data to discover Hamiltonian.”
missing space: “partial differentiations(Rudy”

Are the numbers in the last row of the results table truly exactly the same for Point dynamics and Trajectories (0.33±0.19 0.33±0.18 0.33±0.19 0.33±0.18), or is this an editing error?

> When observing 25-shot point dynamics and 5-shot trajectories, the number of given samples in the phase space is the same, and the same is true for observing 50-shot point dynamics and 10-shot trajectories.
Would be good to refer to L=5 again at this point.

Fig 2 (b-c) I would recommend to add some sort of labels indicating what each of the 4 rows on the left of the plot.

One model seems to be missing from Figure 3(a).

Figure 3(a) caption mentions “after 50 gradient steps” but other places say “10 gradient steps”, not sure if this was changed for Fig 3, or this is a typo.

---

> ### Author Response · Authors · 2020-11-24
> **Response to AnonReviewer1  (part 1)**
>
> First of all, we thank the reviewer’s thoughtful comments and some detailed comments would improve the quality of this paper. Our responses are given as below:
>
>
> 1. Domains are very toy.
>
> In physics and engineering, it is more common and desired how to explain many related systems by a single expression or explanation as simple as possible. But, in most existing studies for learning physical systems, one model is trained per one system, and the evaluations are restricted to the same physical system during training with different initial states. Although the domains we consider are not real-world problems, even in such domains, as shown in our results in experimental sections, the existing model (such as HNN) failed to adapt to new systems. It seems to be limited to be utilized for common physical or engineering problems. Therefore, we focused on learning the shared representation which can be reused to predict new related systems, and modeling many related systems with a single meta-trained model. A similar discussion has been updated in Section 4.1 (For learning ~ observed systems).
>
>
> 2. Gave a hint of what real-world problem settings could benefit from this specific combination.
>
> In Hamiltonian mechanics, the total energy is kinetic energy and potential energy, i.e., H = T + V. For example, the kinetic energy is usually described as $T = \sum p_i^2/{2m_i}$, which is the summation of all kinetic energy of all objects in the system, and the potential energy is given as $V = V(q_1, q_2, q_3, ..., q_n)$. If the well-modeled functional form of the Hamiltonian, researchers can easily predict the dynamics of other systems under the same physical law without a huge amount of data.
> However, it is sometimes hard to model a functional form for unknown processes where it is even uncertain whether a closed-form solution or mathematical expression exists (similar discussion is in Section 1). For example, in many-body physics, the exact form of potential is not always given and the systems are usually imposed to approximate to pairwise-interaction  (i.e., $\sum_{i<j} V(q_i - q_j)$) and held the harmonic potential (i.e. $V(q_i - q_j) = 1/2 m \omega^2 (q_i - q_j)^2$), but the approximations do not hold if the system is in a high energy regime and some other intervention exists. For those cases, our method could be one of the tools for identifying the functional form as a data-driven method.
>
>
> 3. Why MAML is worse than ANIL? More investigations.
>
> We included additional discussion (in Section 2.2 and Section 4.3 in the revised manuscript) to see more supporting evidence about the results, and we conducted additional ablation studies by comparing the learning curves during task-adaptation, and the effects of the size of the meta-train sets (in Section 5.3 Ablation study and Figure 4) where an additional model called HANIL-Inverse (HANIL-INV), which update the parameter except the first layer in the inner-loop, is included. As comparing HAMAML and meta-trained learners with the HANIL-INV and HANIL, the results support the efficacy of the separative learning scheme (MAML update all parameters during the inner-loop, while the others update the subset of the all parameters during the inner-loop), and as the number of updated parameters of HANIL-INV is between those of the HAMAML and HANIL, the results of HANIL-INV behave likewise.

---

> > ### Author Response · Authors · 2020-11-24
> > **Response to AnonReviewer1 (part 2)**
> >
> > 4. Different trends are reported between the original HNN paper and Table 1, especially the results gap between Naive NN (Baseline) and HNN.
> >
> > First, the evaluation settings are very different between the original HNN paper and ours. HNNs have the symplectic structure which is effective prior for learning the Hamiltonian systems having continuous-time symmetry and volume-preserving properties which makes the vector field continuous and conservative. Therefore, HNNs have data efficiency in learning the dynamics in the phase space through the symplectic structures. Meanwhile, in the original HNN paper, the models are trained with a sufficient number of samples (at least hundreds in the original HNN paper) to cover the phase space where the test set is sampled on the same phase space. Therefore, although the HNN is specialized to learn Hamiltonian vector field, Naïve NN also outputs similar values of HNN without aware of the symplectic structure, because Naïve NN also well fit on the given system considered as multi-dimensional regression problem when the sufficient number of samples are given in phase space. In fact, there could be a significant difference between the time-derivative error of Naïve NN and HNN in relatively small data are given to cover phase space (time-derivative error results reported in [1, 2]). In our meta-testing, few samples are given to adapt the learners’ outputs to the correct time-derivatives of the phase space which makes the significant gap between Naïve NN with MAML/ANIL and HMAML/HANIL. Although both Naïve NN with MAML/ANIL and HMANML/HANIL are meta-trained to learned the shared physical laws from related systems, it could be difficult for Naïve NN with MAML/ANIL to predict the appropriate vector field in the state space.
> >
> > Second, the scale of the time-derivatives ($dq/dt$, $dp/dt$) are different. The time-derivative of spring-mass system, pendulum, and Kepler are ($p/m$, $kq$), ($p/ml^2$, $mgl sin{q}$), and ($p/m$, $GMm/q^2$), respectively. As described in Appendix A.1, the physical parameters and the initial states are randomly in the corresponding uniform distributions. Roughly comparing the order of $q$ in $dp/dt$ of each system, $q$, $sin{q}$, $q^{-2}$, it is hard to compare the values of the result of different types of systems on the same line.
> >
> >
> > 5. The title and some sentences are misleading.
> >
> > As the reviewer mention, the main goal of our work is identifying the shared physical laws of related Hamiltonian systems which are the specific instances of the same physical law. To clarify the difference between several/different types of physical systems (such as spring-mass, pendulum, Kepler) and several/new/observed/... systems governed by the same physical law (pendulum systems with different parameters),  we modified the title with ‘Identifying physical law of Hamiltonian systems via meta-learning’, and the phrases representing the different types of systems (such as spring-mass, pendulum, Kepler) have all been corrected to ‘several **types of** physical systems’.
> >
> >
> > 6.	Figure 3 (a) caption mentions “after 50 gradient steps” but other places say “10 gradient steps”, not sure if this was changed for Fig 3, or this is a typo.
> >
> > It is intentional, not a typo. It was difficult to plot the trajectories because some of the trajectories badly diverge after 10 gradient steps. The detailed procedure has been updated and relocated in Section 5.2 Evaluations.
> >
> >
> > 7. Some minor comments:
> >
> > - The trajectories in Figure 3 were sampled at fully random.
> > - We used bold fonts for the best model in Table 1.
> > - We fixed grammar errors.
> > - We added labels indicating each row represent are on the left of the plot in both Figures 2 and 3.
> > - We include the missing trajectories in Figure 3a.
> > - The results for the Kepler problem in table 1 were no editing errors, in fact, all of 0.33s and the corresponding standard deviation indicate different values.
> >
> >
> > [1] Y. Tong, et. al, “Symplectic Neural Networks in Taylor Series Form for Hamiltonian Systems”, arXiv:2005.04986, 2020.
> >
> > [2] T. Matsubara, et. al, “Deep Energy-Based Modeling of Discrete-Time Physics”, in NeurIPS, 2020.

---

> > > ### Comment · AnonReviewer1 · 2020-11-24
> > > **Reviewer response**
> > >
> > > Thanks you for the updated manuscript and the answers, I am reasonably satisfied.
> > >
> > > I am still not fully convinced that the model is novel enough (e.g. simply combines two well known models HNN with MAML ANIL, with the expected conclusions) to grant acceptance, however, since the other reviewers are ok with this, I am happy to increase my score and let the AC make a final decision.

---

### Official Review · AnonReviewer4 · 2020-10-28
**Well written paper with solid contributions**

**Rating:** 7
**Confidence:** 3

**Review:**

The paper presents a meta-learning method for learning Hamiltonian dynamic systems from data. More specifically, the novelty is incorporating Hamiltonian Neural Networks (HNNs) within known meta-learning methods (MAML and ANIL) in order to model new dynamical systems (with previously known structures but unknown parameters) from partially observed data. The results from the experimental evaluation (on three well-known systems) show that such an approach, and in particular HNNs w/ ANIL (HANIL), leads to more accurate models of unseen dynamics compared to other benchmarks methods such as "vanilla" HNNs and HNNs w/ MAML (HAMAML).

Overall, the paper is well written and the contributions are solid: leading to reasonable improvements over recent work on modeling Hamiltonian systems while offering better understanding of the underlying modeling problem. However, I have some concerns regarding the experimental set-up as well as the discussion of related work (and the motivation, thereof), which are reflected in my score.

First, it is unclear to me how the meta-training is performed: The authors state that 10K tasks are being sampled into the meta-training sets -  are these 10K sampled for each of the 3 systems (leading to 30K in total) or 10K in total (what is the task/system distribution in this case)? Is the meta-leaning performed on tasks sampled from all 3 dynamic systems (i.e. one meta-optimization for all 3 systems) or this is done per system (as Fig1 suggests)? The choice of 10K seems a bit arbitrary, how does this choice affects the overall performance? An ablation study on the size of the meta-sets will further highlight the strengths of the proposed method.

Second, the authors chose two benchmarks which address modeling Hamiltonian systems (HNNs) "by design" i.e. vanilla HNNs trained from scratch (for a given task) and Pretrained HNNs (using the meta-sets). Recent work (for instance Symplectic Recurrent Neural Networks (Chen et. al 2020); Hamiltonian Graph NNs (Sanchez-Gonzalez et al. 2019); Symplectic ODE-net (Zhong et al 2020); SympNets (Jin et al 2020)) has shown improvements over the vanilla HNNs both in terms of accuracy and stability. Although these methods are very briefly mentioned in the paper, they are never in-depth discussed nor considered. It would be helpful if (at least) such a discussion is given, as well as maybe discussing possible extensions(eg. SRNN with ANIL).

Next, the authors should consider extending the related work section in few directions. First, since one of the major contributions is related to meta-learning, the authors should provide a related-work segment discussing related meta-learning approaches which would further justify their design choices (eg. why wasn't Reptile considered). Next, in Sec 4.1., the authors state that the "most related" are HNN but omit any further discussion regarding eg. SRNNs (or HGNNs) - why SRNNs are not up-there with HNNs? IMO such a discussion will further motivate the contributions of the paper.

I respectfully disagree with the last statement in Sec 4.2,  - that "In contrast to our work, the existing methods of identifying the governing representations used the symbolic representation underlying the assumptions that the unknown physical laws are expressed by combinations of mathematically known expressions." Namely, the methods in this study also 'implicitly' build upon the very same physical laws (which are combinations of known expressions). The meta-tasks sampling is performed over systems with known structures therefore the very same assumptions still hold.

Other comments:
- Figure 2, while taking a whole page, is a bit unreadable and confusing to follow. More elaborative discussion will help with conveying the message better.
- Can you clarify the difference in the results presented in Table 1 and Figure 3? More specifically, in Figure 3a, the coordinates MSE shows that HAMAML is diverging (while Pre-trained HNNs seem more stable). In Table 1, on the other hand, PreHNN are worse than HAMAML. Are these the same 10 "new" system but at different gradient steps, or?
- In Sec 6, can you clarify this statement "That is because the standard supervised learning scheme is not efficient for the model to learn appropriate shared representation across the systems. "? What is a "standard" supervised scheme and why is not efficient?
- The very next sentence (in Sec 6) states that "Naive NNs ... also fail ... because are hard to grasp the continuous and conservative structure of the vector fields". Can you also clarify this?

----
Post Rebuttal Update:

The authors addressed and clarified many of my concerns, therefore I updated my score.

---

> ### Author Response · Authors · 2020-11-23
> **Response to AnonReviewer4**
>
> First of all, we thank the reviewer’s comments that helped us further highlight our contribution and improve the quality of this paper. Our responses are given as below:
>
> 1. One meta-optimization for all 3 systems? or this is done per system (as Fig 1 suggests)?
>
> 10K sampled tasks are sampled for each of the 3 systems, i.e., 10K pendulums are used for one meta-learning (as Figure 1 suggests). To clarify the difference between several/different types of physical systems (such as spring-mass, pendulum, Kepler) and several/new/observed/... systems governed by the same physical law (pendulum systems with different parameters), phrases representing the meaning of the former have all been replaced by 'several **types of** physical systems'. Also, we modified the title to "Identifying physical law of Hamiltonian systems via meta-learning' to avoid misleading.
>
>
> 2. An ablation study on the size of meta-sets.
>
> We thank the reviewer's comment that reminds us to conduct the ablation study on the effects of the size of tasks. The additional result (in Figure 4b) has been included to show the performance varying as the size of tasks is varying. The result shows that the point at which the effect begins to be noticeable is between 200 and 500 tasks, and slowly converges at around 10,000. More detailed discussions are included in Section 5.3 Ablation study and Figure 4.
>
>
> 3. More discussions for HNNs related works and meta-learning related works.
>
> We have updated related works to the HNNs and highlighted the contributions of our works by comparing the difference between the existing works (in Section 4.1). We also added the gradient-based meta-learning methods and discussed the reason behind the choice of the methods in this paper (in Section 4.3).
>
>
> 4. Disagree with the statement "In contrast to ~ mathematically known expressions." in Sec 4.2.
>
> The statement means that our methods do not utilize any mathematical terms such as $p^2, cos(q)$ for learning Hamiltonian, while the existing symbolic regression methods use the candidate of the mathematical terms for discovering governing equations, which is considered to require relatively strong prior knowledge on the unknown process. However, our method designs the Hamiltonian with simple fully-connected neural networks and using the previous observation of similar systems for learning the meta-transferring representation which is regarded as the physical law. Although the domains we consider in this paper are performed with known structures of Hamiltonian, our method does not access the mathematical expressions of Hamiltonian.
>
>
> 5. For the spring-mass system, the results of Pretrained HNN and HAMAML seem to behave differently in Table 1 and Figure 3.
>
> Table 1 shows the evaluation of adapted learners to 10 new systems and Figure 3 shows one of the examples (i.e., one new system) of the predicted trajectory by the adapted learners. First, the different gradient steps are one reason for the results (10 in Table 1 and 50 in Figure 3). Also, resulting in a large standard deviation in predicting the dynamics of the spring-mass systems as shown in Table 1, Pretrained HNN sometimes yields a more stable result than HAMAML. Since the physical parameters and the initial states in Figure 3 are randomly sampled the seemingly inverse results occurred. For more consistent with Table 1 and easier to distinguish from each other, we re-sampled the trajectories for Figure 3a.
>
>
> 6. What is the 'standard supervised learning scheme'? and why is not efficient?
>
> We took the expression 'standard supervised learning scheme' for meaning 'pretraining a model on all tasks without meta-learning'. For clarifying the vague expression, we have replaced all of the expressions (e.g. 'pretrained across all tasks') in the revised manuscript. A model simply pretrained across all tasks may output the averaged values of time-derivative at each state. As the time-derivatives of each state would varying sensitive to the physical parameters of the systems, the simple averaged values are likely to have very different patterns from the actual vector fields. Therefore, it makes the model less efficient to learn appropriate shared representation than the meta-learned model. The explanation is also added in the revised manuscript.
>
>
> 7. Can you clarify this? "Naive NNs ... also fail ... because are hard to grasp the continuous and conservative structure of the vector fields".
>
> HNNs have the symplectic structure which is effective prior for learning the Hamiltonian systems having continuous-time symmetry and volume-preserving properties which makes the vector field continuous and conservative. As the small number of data are given during meta-testing, Naive NNs fail to grasp such properties without the symplectic structure. Thus, in Figure 2, the phase portraits of Naive NNs are discontinuous or dissipative-like shape.

---

> > ### Comment · AnonReviewer4 · 2020-11-24
> > **response**
> >
> > Thank you for your response. The discussion and the revised manuscript clarified many concerns regarding this work.

---

### Official Review · AnonReviewer2 · 2020-10-29
**Interesting contribution in structure learning of Hamiltonians using Meta-Learning**

**Rating:** 7
**Confidence:** 4

**Review:**

The paper introduces a meta-learning approach in Hamiltonian Neural Networks to find the structure of the Hamiltonian that can be adapted quickly to a new instance of a physical system.

The contribution is novel and the paper is well written. The presentation is mostly clear, however, some improvements are needed.

Strength:
- Clean methodology
- good performance on the task of few-shot learning on a new system
- nice visualization of the vector fields

Weaknesses:
- No real-world problem
- The meta-learner, in particular HANIL, is clearly not adjusting the parameters of the Hamiltonian as we would expect it, like the physical parameters, which would be more at the beginning of the network.  It only adjusts the read-out layer. In this way, it is hard to understand what is really happening.
- You need a dense sampling of the meta-world. Really you have to look at 10000 tasks? A latent variable estimation model would have been a good baseline, or alternative to the Meta-learning framework, for instance, a VAE.

Details:
- below Eq 3: consider calling beta not a learning rate, because it is more a step-size since nothing is really learned in the inner loop
- related work: machine learning methods to perform symbolic regression directly, such as "Learning Equations for Extrapolation and Control"
Sahoo et al, ICML 2018, might be good to add
- Fig 3: font size is far too small, lines are too thin
- Fig 3: consider using a log-scale for the MSE plots

----
Post rebuttal update:
I  read the response and commented on it. The authors clarified my questions and updated the paper accordingly. So I think my score of 7 is supported.

---

> ### Author Response · Authors · 2020-11-23
> **Response to AnonReviewer2**
>
> We thank the reviewer for insightful comments with constructive feedback. Our responses are given as below:
>
>
> 1. No real-world problem.
>
> Although the domains we consider are not real-world problems, even in such domains, as shown in our results in experimental sections, the existing model (such as HNN) failed to adapt to new systems. In physics and engineering, it is common and desired how to explain many related systems by a single expression or even as simple as possible, but it seems to be limited that such existing models to utilized for common physical or engineering problems. Therefore, we focused on verifying whether such meta-learning algorithms are efficient for identifying the physical law of related systems governed by the same physical law. We believe it would be developed to apply to real-world problems in the near future.
>
>
> 2. Hard to understand what is really happening. A latent variable estimation model would have been a good baseline.
>
> We appreciate the reviewer to recommend constructive directions for future works that extend in a way to better understand the physical systems by interpreting what is really happening in network parameters or latent variables. However, in this work, we focused on verifying whether such meta-learning algorithms are efficient for identifying shared representation. We believe that the reviewer's constructive comments would be helpful for our future direction and would be carefully considered how to extend.
>
>
> 3. Really you have to look at 10,000 tasks?
>
> We thank the reviewer's comment that reminds us to conduct the ablation study on the size of tasks. The additional result (Figure 4b in the new manuscript) has been included to show the performance varying as the size of tasks is varying. The result shows that the point at which the effect began to be noticeable is between 200 and 500 tasks, and slowly converges at around 10,000. More detailed discussions are included in Section 5.3 Ablation study.
>
>
> 4. Details:
>
> - We fixed the 'learning rate' to 'step sizes'.
> - We added the recommended related work.
> - We re-plotted Figure 3 with a bigger font, ticker lines, log-scaled MSEs for the spring-mass and pendulum systems, and some additional modifications for better visualization.

---

> > ### Comment · AnonReviewer2 · 2020-11-24
> > **Response**
> >
> > Thank you for your response and the revised version.
> > The ablation for the training set size is helpful.
> >
> > As a hint for your graphics: in matplotlib use figsize=(2,2) or similar (you get big enough axis labels etc.)
> > The legends in fig 3 are still not readable without zooming.

---

> > > ### Author Response · Authors · 2020-11-24
> > > **Figure 3 updated**
> > >
> > > Thank you for your suggestion. We have updated the legends in Figure 3 as bigger fonts.

---

### Author Response · Authors · 2020-11-23
**Revision summary**

We would like to appreciate all the reviewers for constructive comments which have led to the revision and we believe without doubt has improved the quality of the manuscript. The revised manuscript has been uploaded and some major changes are marked with blue color to be more visible. The summary of the major changes is as follow:

1. We have changed the title to "Identifying Physical Law of Hamiltonian Systems via Meta-Learning".

2. We added Hamiltonian neural networks related literature (in Section 4.1), the gradient-based meta-learning methods (in Section 4.3), and the corresponding detailed discussions to highlight the contribution of the paper.

3. We conducted additional ablation studies by comparing the learning curves during task-adaptation, and the effects of the size of the meta-train sets (in Section 5.3 Ablation study and Figure 4).

4. We re-sampled the trajectories of the spring-mass system (in Figure 3 top) and re-plotted the MSEs trajectories of the spring-mass and pendulum systems (in the second and fourth columns of Figure 3 top and middle) as log-scaled for better visualization.

---

### Decision · Program_Chairs · 2021-01-07
**Final Decision**

**Decision:**

Accept (Poster)

**Comment:**

This paper proposes a meta-learning approach for inferring the Hamiltonian governing the dynamics of physical systems from observational data, and using it to adapt to new systems from the same class of dynamics quickly. The paper does this by effectively combining the previously published Hamiltonian Neural Networks and MAML/ANIL. The reviewers agree that the paper is well written, and the experiments are comprehensive, however, they also have reservations about the technical novelty of the proposed solution, given that it appears to be combination of pre-existing models. Saying this, the authors were able to  address a lot of the reviewers' concerns during the discussion period, hence I recommend this paper for acceptance.